# DTWNet: a Dynamic Time Warping Network

**Xingyu Cai**
University of Connecticut

**Tingyang Xu**
Tencent AI Lab

**Jinfeng Yi**
JD.com AI Lab

**Junzhou Huang**
Tencent AI Lab

**Sanguthevar Rajasekaran**
University of Connecticut

## Abstract

Dynamic Time Warping (DTW) is widely used as a similarity measure in various domains. Due to its invariance against warping in the time axis, DTW provides more meaningful discrepancy measurements between two signals than other distance measures. In this paper, we propose a novel component in an artificial neural network. In contrast to the previous successful usage of DTW as a loss function, the proposed framework leverages DTW to obtain a better feature extraction. For the first time, the DTW loss is theoretically analyzed, and a stochastic backpropogation scheme is proposed to improve the accuracy and efficiency of the DTW learning. We also demonstrate that the proposed framework can be used as a data analysis tool to perform data decomposition.

## 1 Introduction

In many data mining and machine learning problems, a proper metric of similarity or distance could play a significant role in the model performance. Minkowski distance, defined as $\text{dist}(x, y) = (\sum_{k=1}^{d} |x_k - y_k|^p)^{1/p}$ for input $x, y \in \mathcal{R}^d$, is one of the most popular metrics. In particular, when $p = 1$, it is called Manhattan distance; when $p = 2$, it is the Euclidean distance. Another popular measure, known as Mahalanobis distance, can be viewed as the distorted Euclidean distance. It is defined as $\text{dist}(x, y) = ((x - y)^T \Sigma^{-1} (x - y))^{1/2}$, where $\Sigma \in \mathcal{R}^{d \times d}$ is the covariance matrix. With geometry in mind, these distance (or similarity) measures, are straightforward and easy to represent.

However, in the domain of sequence data analysis, both Minkowski and Mahalanobis distances fail to reveal the true similarity between two targets. Dynamic Time Warping (DTW) [1] has been proposed as an attractive alternative. The most significant advantage of DTW is its invariance against signal warping (shifting and scaling in the time axis, or Doppler effect). Therefore, DTW has become one of the most preferable measures in pattern matching tasks. For instance, two different sampling frequencies could generate two pieces of signals, while one is just a compressed version of the other. In this case, it will be very dissimilar and deviant from the truth to use the point-wise Euclidean distance. On the contrary, DTW would capture such scaling nicely and output a very small distance between them. DTW not only outputs the distance value, but also reveals how two sequences are aligned against each other. Sometimes, the alignment could be more interesting. Furthermore, DTW could be leveraged as a feature extracting tool, and hence it becomes much more useful than a similarity measure itself. For example, predefined patterns can be identified in the data via DTW computing. Subsequently these patterns could be used to classify the temporal data into categories, e.g., [8]. Some interesting applications can be found in, e.g., [6, 14].

The standard algorithm for computing Dynamic Time Warping involves a Dynamic Programming (DP) process. With the help of $O(n^2)$ space, a cost matrix $C$ would be built sequentially, where

$$C_{i,j} = ||x_i - y_j|| + \min\{C_{i-1,j}, C_{i,j-1}, C_{i-1,j-1}\} \tag{1}$$

Here $||x_i - y_j||$ denotes the norm of $(x_i - y_j)$, e.g., $p$-norm, $p = 1, 2$ or $\infty$. After performing the DP, we can trace back and identify the warping path from the cost matrix. This is illustrated in

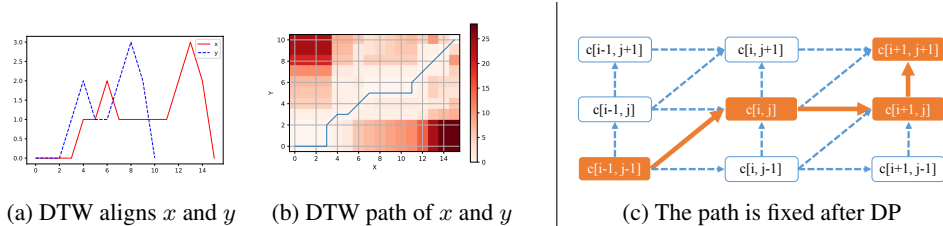

(a) DTW aligns $x$ and $y$      (b) DTW path of $x$ and $y$      (c) The path is fixed after DP

Figure 1: Illustration of DTW Computation, Dynamic Programming and Warping Path

Figure 1a 1b, where two sequences of different lengths are aligned. There are speedup techniques to reduce DTW's time complexity, e.g., [15], which is beyond the scope of this paper. In general, a standard DP requires $O(n^2)$ time.

Although DTW is already one of the most important similarity measures and feature extracting tools in temporal data mining, it has not contributed much to the recent deep learning field. As we know, a powerful feature extractor is the key to the success of an artificial neural network (ANN). The best example could be the CNNs that utilize convolutional kernels to capture local and global features [10]. Unlike the convolution, DTW has the non-linear transformation property (warping), providing a summary of the target against Doppler effects. This makes DTW a good candidate as a feature extractor in general ANNs. With this motivation, we propose DTWNet, a neural network with learnable DTW kernels.

**Key Contributions:** We apply the learnable DTW kernels in neural networks to represent Doppler invariance in the data. To learn the DTW kernel, a stochastic backpropagation method based on the warping path is proposed, to compute the gradient of a DP process. A convergence analysis of our backpropagation method is offered. To the best of the authors' knowledge, for the first time, DTW loss function is theoretically analyzed. A differentiable streaming DTW learning is also proposed to overcome the problem of missing local features, caused by global alignment of the standard DTW. Empirical study shows the effectiveness of the proposed backpropagation and the success of capturing features using DTW kernels. We also demonstrate a data decomposition application.

## 2 Related Work

### 2.1 Introduction of Dynamic Time Warping

Dynamic Time Warping is a very popular tool in temporal data mining. For instance, DTW is invariant of Doppler effects thus it is very useful in acoustic data analysis [14]. Another example is that biological signals such as ECG or EEG, could use DTW to characterize potential diseases [24]. DTW is also a powerful feature extractor in conjunction with predefined patterns (features), in the time series classification problem [8]. Using Hamming distance, the DTW alignment in this setting is called the Edit distance and also well studied [7].

Due to the Dynamic Programming involved in DTW computation, the complexity of DTW can be high. More critically, DP is a sequential process which makes DTW not parallelizable. To speedup the computation, some famous lower bounds based techniques [9, 12, 20] have been proposed. There are also attempts on parallelization of DP [25] or GPU acceleration [22].

Two dimensional DTW has also drawn research interests. In [11], the author showed that the DTW could be extended to the 2-D case for image matching. Note that this is different from another technique called **multi-variate DTW** [23, 13, 21], sometimes also referred to as multi-dimensional DTW. In multi-variate DTW, the input is a set of 1-D sequences, e.g., of dimension $k \times n$ where $n$ is the sequence length. However, in 2-D or $k$-D DTW, the input is no longer a stack of 1-D sequences but images ($n^2$) or higher dimensional volumes ($n^k$). As a result, the cost of computing 2-D DTW can be as high as $O(n^6)$ and thus making it not applicable for large datasets.

### 2.2 SPRING Algorithm, the Streaming Version of DTW

To process the streaming data under DTW measure, [18] proposed a modified version of DTW computation called SPRING. The original DTW aims to find the best alignment between two input sequences, and the alignment is from the beginning of both sequences to the end. On the contrary,

the streaming version tries to identify all the subsequences from a given sequence, that are close to a given pattern under the DTW measure. The naive approach computes DTW between all possible subsequences and the pattern. Let the input sequence and the pattern be of lengths $n$ and $l$, respectively. The naive method takes $(nl + (n-1)l + \ldots) = O(n^2 l)$ time. However, SPRING only takes $O(nl)$ time, which is consistent with the standard DTW.

SPRING modifies the original DTW computation with two key factors. First, it prepends one wild-card to the pattern. When matching the pattern with the input, since the wild-card can represent any value, the start of the pattern could match any position in the input sequence at no cost. The second modification is that SPRING makes use of an auxiliary matrix to store the source of each entry in the original dynamic programming matrix. This source matrix will keep records of each candidate path and hence we can trace back from the end. Interested readers could refer to [18] for more details.

## 2.3 DTW as a Loss Function

Recently, in order to apply the DTW distance for optimization problems, the differentiability of DTW has been discussed in the literature. As we know, computing DTW is a sequential process in general. During the filling of the DP matrix, each step takes a min operation on the neighbors. Since the min operator is not continuous, the gradient or subgradient is not very well defined. The first attempt to use soft-min function to replace min is reported in [19]. In their paper, the authors provide the gradient of soft-min DTW, and perform shapelet learning to boost the performance of time series classification in limited test datasets. Using the same soft-min idea, in [4], the authors empirically show that applying DTW as a loss function leads to a better performance than conventional Euclidean distance loss, in a number of applications. Another very recent paper [2] also uses continuous relaxation of the min operator in DTW to solve video alignment and segmentation problems.

## 3 Proposed DTW Layer and its Backpropogation

In this paper, we propose to use DTW layers in a deep neural network. A DTW layer consists of multiple DTW kernels that extract meaningful features from the input. Each DTW kernel generates a single channel by performing DTW computation between the kernel and the input sequences. For regular DTW, one distance value will be generated for each kernel. For the streaming DTW, multiple values would be output (details will be given in § 5). If using a sliding window, the DTW kernel would generate a sequence of distances, just as a convolutional kernel. After the DTW layer, linear layers could be appended, to obtain classification or regression results. A complete example of DTWNet on a classification task is illustrated in Algorithm 1.

---

**Algorithm 1** DTWNet training for a classification task. Network parameters are: number of DTW kernels $N_{\text{kernel}}$; kernels $x_i \in \mathcal{R}^l$; linear layers with weights $w$.

---

**INPUT:** Dataset $Y = \{(y_i, z_i)|y_i \in \mathcal{R}^n, z_i \in \mathcal{Z} = [1, N_{\text{class}}]\}$. The DTWNet dataflow can be denoted as $\mathcal{G}_{x,w} : \mathcal{R}^n \to \mathcal{Z}$.
**OUTPUT:** The trained DTWNet $\mathcal{G}_{x,w}$
1: Init $w$; For $i = 1$ to $N_{\text{kernel}}$: randomly init $x_i$; Set total # of iteration be $T$, stopping condition $\epsilon$
2: **for** $t = 0$ to $T$ **do**
3:     Sample a mini-batch $(y, z) \in Y$. Compute DTWNet output: $\hat{z} \leftarrow \mathcal{G}_{x,w}(y)$
4:     Record warping path $\mathcal{P}$ and obtain determined form $f_t(x, y)$, as in Equation 2
5:     Let $\mathcal{L}_t \leftarrow \mathcal{L}_{CrossEntropy}(\hat{z}, z)$. Compute $\nabla_w \mathcal{L}_t$ through regular BP.
6:     For $i = 1$ to $N_{\text{kernel}}$: compute $\nabla_{x_i}\mathcal{L}_t \leftarrow \nabla_{x_i} f_t(x_i, y)\frac{\partial \mathcal{L}_t}{\partial f_t}$ based on $\mathcal{P}$, as in Equation 3
7:     SGD Update: let $w \leftarrow w - \alpha\nabla_w \mathcal{L}_t$ and for $i = 1$ to $N_{\text{kernel}}$ do $x_i \leftarrow x_i - \beta\nabla_{x_i}\mathcal{L}_t$
8:     If $\Delta\mathcal{L} = |\mathcal{L}_t - \mathcal{L}_{t-1}| < \epsilon$: return $\mathcal{G}_{x,w}$

---

**Gradient Calculation and Backpropogation**

To achieve learning of the DTW kernels, we propose a novel gradient calculation and backpropogation (BP) approach. One simple but important observation is that: after performing DP and obtaining the warping path, the path itself is settled down for this iteration. If the input sequences and the kernel are of lengths $n$ and $l$, respectively, the length of the warping path cannot be larger than $O(n + l)$.

This means that the final DTW distance could be represented using $O(n + l)$ terms, and each term is $||y_i - x_j||$ where $i, j \in S$, and $S$ is the set containing the indices of elements along the warping path. For example, if we use 2-norm, the final squared DTW distance could be of the following form:

$$\text{dtw}^2(x, y) = f_t(x, y) = ||y_0 - x_0||_2^2 + ||y_1 - x_0||_2^2 + ||y_2 - x_1||_2^2 + \ldots \qquad (2)$$

This is illustrated in Figure 1c, where the solid bold lines and the highlighted nodes represent the warping path after Dynamic Programming. Since the warping path is determined, other entries in the cost matrix no longer affect the DTW distance, thus the differentiation can be done only along the path. Since the DTW distance obtains its determined form, e.g., Equation 2, taking derivative with respect to either $x$ or $y$ becomes trivial, e.g.,

$$\nabla_x \text{dtw}^2(x, y) = \nabla_x f_t(x, y) = [2(y_0 + y_1 - 2x_0) , \ 2(y_2 - x_1) , \ \ldots]^T \qquad (3)$$

Since the min operator does not have a gradient, directly applying auto-diff will result in a very high variance. Soft-min could somewhat mitigate this problem, however, as shown above, since the final DTW distance is only dependent on the elements along the warping path, differentiation on all the entries in the cost matrix becomes redundant. Other than this, additional attention needs to be paid to the temperature hyperparameter in the soft-min approach, which controls the trade-off between accuracy and numerical stability.

In contrast, taking derivative using the determined form along the warping path, we can avoid the computation redundancy. As the warping path length cannot exceed $O(n + l)$, the differentiation part only takes $O(n + l)$ time instead of $O(nl)$ as in the soft-min approaches. Note that there is still a variance which arises from the difference in DP's warping paths from iteration to iteration, so the BP can be viewed as a stochastic process.

**Time Complexity:** The computation of DTW loss requires building a Dynamic Programming matrix. The standard DP needs $O(nl)$ time. There are speeding-up/approximating techniques for DP such as banded constraint (limit the warping path within a band), which is beyond the scope of this paper. The gradient is evaluated in $O(n + l)$ time as shown above. Although the DP part is not parallelizable in general, parallelization can still be achieved for independent evaluation for different kernels.

## 4 DTW Loss and Convergence

To simplify the analysis, we consider that for one input sequence $y \in \mathcal{R}^n$. The goal is to obtain a target kernel $x \in \mathcal{R}^l$ that has the best alignment with $y$, i.e., $\min_x \text{dtw}^2(x, y)$. Without loss of generality, we assume $l \leq n$. The kernel $x$ is randomly initialized and we perform learning through standard gradient descent. Define the DTW distance function as $d = H_y(x)$, where $d \in \mathcal{R}$ is the DTW distance evaluated by performing the Dynamic Programming operator, i.e., $d = \text{DP}(x, y)$.

**Definition 1.** *Since DP provides a deterministic warping path for arbitrary $x$, we define the space of all the functions of $x$ representing all possible warping paths as*

$$\mathcal{F}_y = \{f_y(x) | f_y(x) = \sum_{i,j} I_{ij} ||(x_i - y_j)||_2^2\}$$

$$s.t. \quad i \in [0, l - 1]; \ j \in [0, n - 1]; \ I_{ij} \in \{0, 1\}; \ n \leq |I| \leq n + l;$$
$$i, j \text{ satisfy temporal order constraints.}$$

Here the cardinality of $I$ is within the range of $n$ and $n + l$, because the warping path length can only be between $n$ and $n + l$. The temporal order constraints make sure that the combination of $i, j$ must be valid. For example, if $x_i$ is aligned with $y_j$, then $x_{i+1}$ cannot be aligned with $y_{j-1}$, otherwise the alignment will be against the DTW definition.

With Definition 1, when we perform Dynamic Programming at an arbitrary point $x$ to evaluate $H_y(x)$, we know that it must be equal to some function sampled from the functional space $\mathcal{F}_y$, i.e., $H_y(x)|_{x=\hat{x}} = f_y^{(u)}(x)|_{x=\hat{x}}$, $f_y^{(u)} \in \mathcal{F}_y$. So we can approximate $H_y(x)$ as a collection of functions in $\mathcal{F}_y$, where each $x$ could correspond to its own sample function. In the proposed backpropagation step we compute the gradient of $f_y^{(u)}(x)$ and perform the gradient descent using this gradient. The first question is whether $\nabla_x f_y^{(u)}(x)|_{x=\hat{x}} = \nabla_x H_y(x)|_{x=\hat{x}}$.

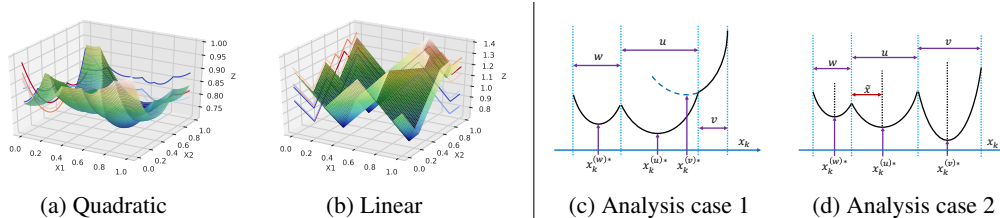

| (a) Quadratic | (b) Linear | (c) Analysis case 1 | (d) Analysis case 2 |

Figure 2: Loss function $d = H_y(x)$ and analysis. (a): $H_y(x)$ approximated by quadratic $f_y(x)$; (b): by linear $f_y(x)$; The curves on the wall are projections of $H_y(x)$ for better illustration. (c): Illustration of transitions from $u$ to $v$, here $f_y^{(v)}$'s stationary point (where $\nabla_{x_k} f_y^{(v)} = 0$) is outside of $v$; (d): both $u$ and $v$ have bowl-shapes.

We notice the fact that $H_y(x)$ is not smooth in the space of $x$. More specifically, there exist positions $x$ such that

$$H_y(x) = \begin{cases} f_y^{(u)}(x)|_{x=x_+} \\ f_y^{(v)}(x)|_{x=x_-} \end{cases} \quad u \neq v; \ f_y^{(u)}, f_y^{(v)} \in \mathcal{F}_y \tag{4}$$

where $x_+$ and $x_-$ represent infinitesimal amounts of perturbation applied on $x$, in the opposite directions. However, note that the cardinality of $\mathcal{F}_y$ is finite. In fact, in the Dynamic Programming matrix, for any position, the warping path can only evolve in at most three directions, due to the temporal order constraints. In boundary positions, only one direction can the warping path evolve along. So we have:

**Lemma 1.** *Warping paths number* $|\mathcal{F}_y| < 3^{n+l}$, *where* $(n + l)$ *is the largest possible path length.*

This means that the space of $x$ is divided into regions such that $H_y(x)$ is perfectly approximated by $f_y^{(u)}(x)$ in the particular region $u$. In other words, the loss function $H_y(x)$ is a piece-wise (or region-wise) quadratic function of $x$, if we compute the DTW loss as a summation of squared 2-norms, e.g., $\text{dtw}^2(x, y) = ||x_0 - y_0||_2^2 + ||x_1 - y_0||_2^2 + \dots$. Similarly, if we use the absolute value as the element distance for the functions in the set $\mathcal{F}_y$, then we obtain piece-wise linear function as $H_y(x)$.

This is shown in Figure 2a, 2b. We perform Monte-Carlo simulations to generate the points and compute their corresponding DTW loss. The length of $x$ is 6, but we only vary the middle two elements after a random initialization and hence can generate the 3-D plots. The length of $y$ is 10. The elements in both $x$ and $y$ are randomly initialized within $[0, 1]$. Figure 2a verifies that $H_y(x)$ is piece-wise quadratic using 2-norms, where Figure 2b corresponds to the piece-wise linear function.

**Escaping Local Minima**

Some recent theoretical work provides proofs for global convergence in non-convex neural network loss functions, e.g., see [5]. In this paper, we offer a different perspective for the analysis by exploiting the fact that the global loss function is piece-wise quadratic or linear obtained by a DP process, and the number of regions is bounded by $O(3^{n+l})$ (Lemma 1). Without loss of generality, we only consider $H_Y(x)$ being piece-wise quadratic. Treating the regions as a collection of discrete states $V$, where $|V| < 3^{n+l}$, we first analyze the behavior of escaping $u$ and jumping to its neighbor $v$, for $u, v \in V$, using the standard gradient descent. Without loss of generality, we only look at coordinate $k$ ($x_k$ is of interest). Assume that after DP, a fraction $y_{p:p+q}$ is aligned with $x_k$. Taking out the items related to $x_k$, we can write the local quadratic function in $u$, and its partial derivative with respect to $x_k$, as

$$f_y^{(u)} = \sum_{j=p}^{p+q} (y_j - x_k)^2 + \sum_{i,j \in \mathcal{U}} I_{ij}(x_i - y_j)^2 \quad \text{and} \quad \nabla_{x_k} f_y^{(u)} = \sum_{j=p}^{p+q} 2(x_k - y_j) \tag{5}$$

where $\mathcal{U} = \{i, j | i \neq k, j \notin [p, p+q]\}$, $I_{ij} \in \{0, 1\}$, which is obtained through DP, and $i, j$ satisfy temporal order. Setting $\nabla_{x_k} f_y^{(u)} = 0$ we get the stationary point at $x_k^{(u)*} = \frac{1}{q+1} \sum_{j=p}^{p+q} y_j$.

Without loss of generality, consider the immediate neighbor $f_y^{(v)}$, the same as $f_y^{(u)}$ except for only the alignment of $y_{p+q+1}$, i.e.,

$$f_y^{(v)} = \sum_{j=p}^{p+q+1} (y_j - x_k)^2 + \sum_{i,j \in \mathcal{V}} I_{ij}(x_i - y_j)^2 \tag{6}$$

where $\mathcal{V} = \{i, j | i \neq k, j \notin [p, p+q+1]\}$. The corresponding stationary point is at $x_k^{(v)*}$. Similarly, for the other immediate neighbor $w$ that aligns $\sum_{j=p}^{p+q-1} y_j$, the stationary point is at $x_k^{(w)*}$. We have

$$x_k^{(u)*} = \frac{\sum_{j=p}^{p+q} y_j}{q+1} \quad, \quad x_k^{(v)*} = \frac{\sum_{j=p}^{p+q+1} y_j}{q+2} \quad, \quad x_k^{(w)*} = \frac{\sum_{j=p}^{p+q-1} y_j}{q} \tag{7}$$

Without loss of generality, assume that the three neighbor regions $w, u, v$ are from left to right, i.e., $x_k^1 < x_k^2 < x_k^3$, for $x_k^1 \in w, x_k^2 \in u, x_k^3 \in v$. The three regions corresponding to three local quadratic functions $f_y^{(w)}, f_y^{(u)}, f_y^{(v)}$, and their local minima (or stationary points) $x_k^{(w)*}, x_k^{(u)*}, x_k^{(v)*}$, are illustrated in Figure 2c, 2d. Note that we are interested in transition $u \to v$, when $u$'s local minimum is not at the boundary ($u$ has a bowl-shape and we want to jump out).

There could be 3 possibilities for the destination (region $v$). The first one is illustrated in Figure 2c, where $x_k^{(v)*}$ is not inside region $v$, but somewhere to the left. In this case, it is easy to see the global minimum will not be in $v$ since some part in $u$ is lower ($u$ has the bowl-shape due to its local minimum). If jumping to $v$, the gradient in $v$ would point back to $u$, which is not the case of interest.

In the second case, both $u$ and $v$ have the bowl-shapes. As shown in Figure 2d, the distance between the bottom of two bowls is $d_k^{(u,v)} = x_k^{(v)*} - x_k^{(u)*}$. The boundary must be somewhere in between $x_k^{(u)*}$ and $x_k^{(v)*}$. Since we need to travel from $u$ to $v$, the starting point $x_k = \tilde{x} \in u$ must be to the left of $x_k^{(u)*}$ (as shown in the red double-arrows region, in Figure 2d). Otherwise the gradient at $\tilde{x}$ will point to region $w$ instead of $v$. To ensure one step crossing the boundary and arrives at $v$, it needs to travel a distance of at most $(x_k^{(v)*} - \tilde{x})$, because the boundary between $u$ and $v$ could never reach $x_k^{(v)*}$.

For the third case, $v$ does not have the bowl-shape, but $x_k^{(v)*}$ is to the right of $v$. We can still travel $(x_k^{(v)*} - \tilde{x})$ to jump beyond $v$. Similar to case 1, the right neighbor of $v$ (denoted as $v+$) would have a lower minimum if $v+$ has bowl-shape. Even if $v+$ does not have a bowl-shape, the combined region $[v, v+]$ can be viewed as either a quasi-bowl or an extended $v$, thus jumping here is still valid.

Next, we need to consider the relationship between feasible starting point $\tilde{x}$ and $f_y^{(w)}$'s stationary point $x_k^{(w)*}$. If $x_k^{(w)*}$ is within region $w$, since $\tilde{x} \in u$, we know that $\tilde{x} > x_k^{(w)*}$. However, there could be cases in which $w$ does not hold $f_y^{(w)}$'s stationary point. If the stationary point $x_k^{(w)*}$ is to the left of region $w$, then the inequality $\tilde{x} > x_k^{(w)*}$ becomes looser, but still valid. Another case is that when $x_k^{(w)*}$ is to the right side of $w$. This means $w$ is monotonically decreasing, so we can combine $[w, u]$ as a whole quasi-bowl region $u'$, and let $w'$ be the left neighbor of the combined $u'$. Therefore, the above analysis on $w', u'$ and $v$ still holds, and we want to jump out $u'$ to $v$. Hence we arrive at the following theorem.

**Theorem 1.** *Assume that the starting point at coordinate $k$, i.e. $x_k = \tilde{x}$, is in some region $u$ where $f_y^{(u)}$ is defined in Equation 5. Let $x$ and $y$ have lengths $n$ and $l$, respectively, and assume that $l < n$. To ensure escaping from $u$ to its immediate right-side neighbor region, the expected step size $\mathbb{E}[\eta]$ needs to satisfy: $\mathbb{E}[\eta] > \frac{l}{2n}$.*

The proof can be found in the supplementary A. In other cases, we consider a dataset $Y = \{y_i | y_i \in \mathcal{R}^n, i = 1, \ldots, m\}$. The DTW loss and its full gradient have the summation form, i.e., $H_Y(x) = \sum_{i=0}^m H_{y_i}(x)$ and $\nabla_x H_Y(x) = \sum_{i=0}^m \nabla_x H_{y_i}(x)$. The updating of $x$ is done via stochastic gradient descent (SGD) over mini-batches, i.e., $x \leftarrow x + \eta \frac{m}{b} \sum_b \nabla_x H_{y_i}(x)$, where $b < m$ is the mini-batch size, and $\eta$ is the step size. Though the stochastic gradient is an unbiased estimator, i.e., $\mathbb{E}[\frac{m}{b} \sum_b \nabla_x H_{y_i}(x)] = \nabla_x H_Y(x)$, the variance offers the capability to jump out of local minima.

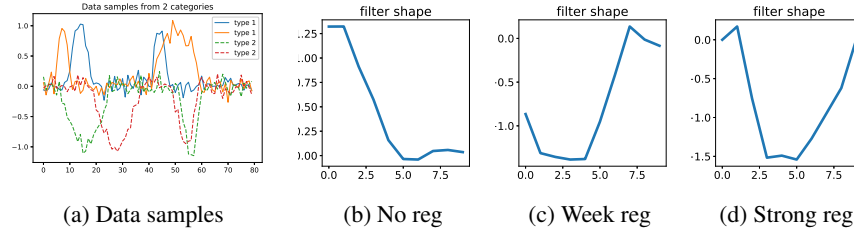

(a) Data samples      (b) No reg      (c) Week reg      (d) Strong reg

Figure 3: Illustration of the effect of the streaming DTW's regularizer: from left to right, $\alpha = 0$ and $1 \times 10^{-4}$ and 0.1, respectively.

## 5 Streaming DTW Learning

The typical length of a DTW kernel is much shorter than the input data. Aligning the short kernel with a long input sequence, could lead to misleading results. For example, consider the ECG data sequence which consists of several periods of heartbeat pulses, and we would like to let the kernel learn the heartbeat pulse pattern. However, applying an end-to-end DTW, the kernel will align the entire sequence rather than a single pulse period. If the kernel is very short, it does not even have enough resolution and thus finally outputs a useless abstract.

To address this problem, we bring the SPRING [18] algorithm to output the patterns aligning subsequences of the original input:

$$x^* = \arg \min_{i,\Delta,x} \text{dtw}^2(x, y_{i:i+\Delta}) \tag{8}$$

where $y_{i:i+\Delta}$ denotes the subsequence of $y$ that starts at position $i$ and ends at $i + \Delta$, and $x$ is the pattern (the DTW kernel) we would like to learn. Note that $i$ and $\Delta$ are parameters to be optimized.

In fact, SPRING not only finds the best matching among all subsequences, but also reports a number of candidate warping paths that have small DTW distances. As a result, we propose two schemes that exploit this property. In the first scheme, we pre-specify a constant $k$ (e.g. 3 or 5) and let SPRING provide the top $k$ best warping paths ($k$ different non-overlapping subsequences that have least DTW distances to the pattern $x$). In the second scheme, rather than specifying the number of paths, we set a value of $\epsilon$ such that all the paths that have distances smaller than $(1 + \epsilon)d^*$ are reported, where $d^*$ is the best warping path's DTW distance. After obtaining multiple warping paths, we can do either an averaging, or random sampling as our DTW computing result. In our experiments, we choose $\epsilon = 0.1$ and randomly sample one path for simplicity.

**Regularizer in Streaming DTW**

Since SPRING encourages the kernel $x$ to learn some repeated pattern in the input sequence, there is no constraint of such patterns' shapes, which could cause problematic learning results. As a matter of fact, some common shapes that do not carry much useful information always occur in the input data. For example, an up-sweep or down-sweep always exists, even the Gaussian noise is a combination of such sweeps. The kernel without any regularization would easily capture such useless patterns and fall into such local minima. To solve this issue, we propose a simple solution that adds a regularizer on the shape of the pattern. Assuming $x$ is of length $l$, we change the objective to

$$\min_{i,\Delta,x} (1-\alpha)\text{dtw}^2(x, y_{i:i+\Delta}) + \alpha||x_0 - x_l|| \tag{9}$$

where $\alpha$ is the hyper parameter that controls the regularizer. This essentially forces the pattern to be a "complete" one, in the sense that the beginning and the ending of the pattern should be close. It is a general assumption that we want to capture such "complete" signal patterns, rather than parts of them. As shown in Figure 3a, the input sequences contain either upper or lower half circles as the target to be learned. Without regulation, Figure 3b shows that the kernel only learns a part of that signal. Figure 3c corresponds to a weak regularizer, where the kernel tries to escape from the tempting local minima (these up-sweeps are so widely spread in the input and lead to small SPRING DTW distances). A full shape is well learned with a proper $\alpha$, as shown in Figure 3d. Other shape regularizers could be also used, if they contain prior knowledge from human experts.

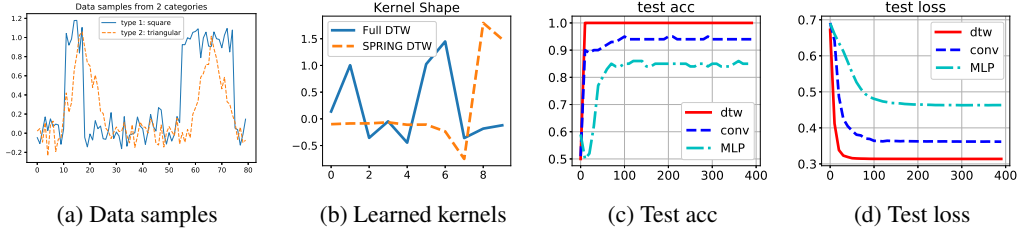

Figure 4: Performance comparison on synthetic data sequences (400 iterations)

# 6 Experiments and Applications

In this experimental section, we compare the proposed scheme with existing approaches. We refer to the end-to-end DTW kernel as **Full DTW**, and the streaming version as **SPRING DTW**. We implement our approach in PyTorch [16].

## 6.1 Comparison with Convolution Kernel

In this very simple classification task, two types of synthetic data sequences are generated. Category 1 only consists of half square signal patterns. Category 2 only has upper triangle signal patterns. Each data sequence was planted with two such signals, but in random locations with random pattern lengths. The patterns do not overlap in each sequence. Also, Gaussian noise is injected into the sequences. Figure 4a provides some sample sequences from both categories.

The length of the input sequences is 100 points, where the planted pattern length varies from 10 to 30. There are a total of 100 sequences in the training set, 50 in each category. Another 100 sequences form the testing set, 50 for each type as well. We added Gaussian noise with $\sigma = 0.1$. For comparison, we tested one full DTW kernel, one SPRING DTW kernel, and one convolution kernel. The kernel lengths are set to 10. $\alpha = 0.1$ for SPRING DTW. We append 3 linear layers to generate the prediction.

In Figure 4b, we show the learned DTW kernel after convergence. As expected, the full DTW kernel tries to capture the whole sequence. Since the whole sequence consists of two planted patterns, the full DTW also has two peaks. On the contrary, SPRING DTW only matches partial signal, thus resulting in a sweep shape. Figure 4c and Figure 4d show the test accuracy and test loss for 400 iterations. Since both full DTW and SPRING DTW achieve 100% accuracy, and their curves are almost identical, we only show the curve from the full DTW. Surprisingly, the network with the convolution kernel fails to achieve 100% accuracy after convergence on this simple task. The "MLP" represents a network consisting of only 3 linear layers, and performs the worst among all the candidates as expected.

Note that we can easily extend the method to multi-variate time series data (MDTW [21]), without any significant modifications. Details can be found in the supplementary B.

## 6.2 Evaluation of Gradient Calculation

To evaluate the effectiveness and accuracy of the proposed BP scheme, we follow the experimental setup in [4] and perform barycenter computations. The UCR repository [3] is used in this experiment. We evaluate our method against SoftDTW [4], DBA [17] and SSG [4]. We report the average of 5 runs for each experiment. A random initialization is done for all the methods. Due to space limit, we only provide a summary in this section but details can be found in supplementary C (Table 2, 3).

The barycenter experiment aims to find the barycenter for the given input sequences. We use the entire training set to train the model to obtain the barycenter $b_i$ for each category, and then calculate the DTW loss as:

$$\mathcal{L}_{\text{dtw}} = \frac{1}{N_{\text{class}}} \sum_{i=0}^{N_{\text{class}}} \frac{1}{N_i} \sum_{j=0}^{N_i} \text{dtw}(s_{i,j}, b_i) \tag{10}$$

where $N_{\text{class}}$ is the number of categories, $N_i$ is the number of sequences in class $i$, and $s_{i,j}$ is sequence $j$ in class $i$. The DTW distance is computed using $\ell_2$ norm. Clearly, the less the loss, the better is the

Table 1: Barycenter Experiment Summary

| Alg | Training Set | | | | Testing Set | | | |
|---|---|---|---|---|---|---|---|---|
| | SoftDTW | SSG | DBA | **Ours** | SoftDTW | SSG | DBA | **Ours** |
| Win | 4 | 23 | 21 | **37** | 11 | 21 | 22 | **31** |
| Avg-rank | 3.39 | **2.14** | 2.27 | 2.2 | 3.12 | 2.31 | 2.36 | **2.21** |
| Avg-loss | 27.75 | 26.19 | 26.42 | **24.79** | 33.08 | 33.84 | 33.62 | **31.99** |

performance. We also evaluate on the testing set by using $s_{i,j}$ from the testing set. Note that we first run SoftDTW with 4 different hyperparameter settings $\gamma = 1, 0.1, 0.01$ and $0.001$ as in [4]. In the training set, $\gamma = 0.1$ outperforms others, while in the testing set, $\gamma = 0.001$ gives the best results, thus we select $\gamma$ accordingly.

The experimental results are summarized in Table 1. "Win" denotes the number of times the smallest loss was achieved, among all the 85 datasets. We also report the average rank and average loss (sum all the losses and divide by number of datasets) in the table. From the results we can clearly see that our proposed approach achieves the best performance among these methods. The details of this experiment can be found in supplementary C.

### 6.3 Application of DTW Decomposition

In this subsection, we propose an application of DTWNet as a time series data decomposition tool. Without loss of generality, we design 5 DTW layers and each layer has one DTW kernel, i.e., $x_i$. The key idea is to forward the residual of layer $i$ to the next layer in this network. Note that DTW computation $\text{dtw}(y, x_i)$ will generate the warping path like Equation 2, from which we obtain the residual by subtracting the corresponding aligned $x_{i,j}$ from $y_j$, where $j$ is the index of elements.

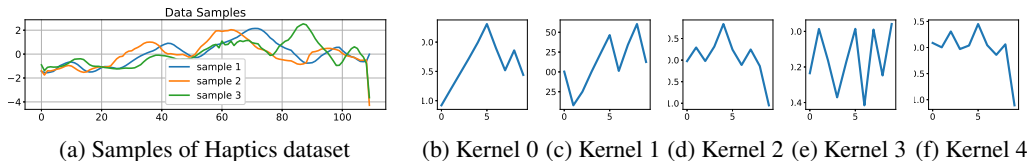

(a) Samples of Haptics dataset    (b) Kernel 0 (c) Kernel 1 (d) Kernel 2 (e) Kernel 3 (f) Kernel 4

Figure 5: Illustration of DTW Decomposition

Figure 5 illustrates the effect of the decomposition. Kernel $0$ to kernel $4$ correspond to the first layer (input side) till the last layer (output side). The training goal is to minimize the residual of the network's output, and we randomly initialize the kernels before training. We use the Haptics dataset from the UCR repository to demonstrate the decomposition.

After a certain amount of epochs, we can clearly see that the kernels from different layers form different shapes. The kernel 0 from the first layer, has a large curve that describes the overall shape of the data. This can be seen as the low-frequency part of the signal. In contrast, kernel 4 has those zig-zag shapes that describe the high-frequency parts. Generally, in deeper layers, the kernels tend to learn "higher frequency" parts. This can be utilized as a good decomposition tool given a dataset. More meaningfully, the shapes of the kernels are very interpretable for human beings.

## 7 Conclusions and Future Work

In this paper, we have applied DTW kernel as a feature extractor and proposed the DTWNet framework. To achieve backpropagation, after evaluating DTW distance via Dynamic Programming, we compute the gradient along the determined warping path. A theoretical study of the DTW as a loss function is provided. We identify DTW loss as region-wise quadratic or linear, and describe the conditions for the step size of the proposed method in order to jump out of local minima. In the experiments, we show that the DTW kernel could outperform standard convolutional kernels in certain tasks. We have also evaluated the effectiveness of the proposed gradient computation and backpropagation, and offered an application to perform data decomposition.

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
