[Supplementary Material]

# A Proof of Theorem 1

*Proof.* As discussed above, we have two cases as follows.

**Case 1:** First we consider the case that $w, u, v$ are from left to right, with their stationary points $x_k^{(w)*} < x_k^{(u)*} < x_k^{(v)*}$. A standard gradient descent update is $\tilde{x} \leftarrow \tilde{x} - \eta \nabla_{x_k} f_y^{(u)}|_{x_k=\tilde{x}}$. To ensure one step update could make $\tilde{x}$ jump from $u$ to $v$, we obtain:

$$
\begin{aligned}
&\tilde{x} - \eta \nabla_{x_k} f_y^{(u)}|_{x_k=\tilde{x}} \geq x_k^{(v)*} \\
\Rightarrow\ &\tilde{x} - \eta \sum_{j=p}^{p+q} 2(\tilde{x} - y_j) \geq x_k^{(v)*} \text{ (use Equation 5)} \\
\Rightarrow\ &(1 - 2\eta(q+1))\tilde{x} + 2\eta(q+1)\frac{1}{q+1}\sum_{j=p}^{p+q} y_j \geq x_k^{(v)*} \\
\Rightarrow\ &(1 - 2\eta(q+1))\tilde{x} \geq x_k^{(v)*} - 2\eta(q+1)x_k^{(u)*} \text{ (use Equation 7)}
\end{aligned}
\tag{11}
$$

Now we have two possibilities:

**Possibility (a):** $1 - 2\eta(q+1) > 0$:

$$
\text{Inequality 11} \Rightarrow \tilde{x} \geq \frac{x_k^{(v)*} - 2\eta(q+1)x_k^{(u)*}}{1 - 2\eta(q+1)}
$$

To guarantee the gradient direction points to neighbor $v$, the starting point $\tilde{x}$ has to be to the left of $x_k^{(u)*}$, thus:

$$
\begin{aligned}
&\frac{x_k^{(v)*} - 2\eta(q+1)x_k^{(u)*}}{1 - 2\eta(q+1)} \leq \tilde{x} < x_k^{(u)*} \\
\Rightarrow\ &x_k^{(v)*} - 2\eta(q+1)x_k^{(u)*} < (1 - 2\eta(q+1))x_k^{(u)*} \\
\Rightarrow\ &x_k^{(v)*} < x_k^{(u)*}
\end{aligned}
\tag{12}
$$

This is contradictory to the assumption that $x_k^{(w)*} < x_k^{(u)*} < x_k^{(v)*}$, and thus is not valid.

**Possibility (b):** $1 - 2\eta(q+1) < 0$:

$$
\text{Inequality 11} \Rightarrow \tilde{x} \leq \frac{x_k^{(v)*} - 2\eta(q+1)x_k^{(u)*}}{1 - 2\eta(q+1)}
$$

Due to the fact that $\tilde{x}$ is inside region $u$, which must be somewhere to the right of $x_k^{(w)*}$ (with the assumption that $w$ has a bowl-shape), we have:

$$
\begin{aligned}
&\frac{x_k^{(v)*} - 2\eta(q+1)x_k^{(u)*}}{1 - 2\eta(q+1)} \geq \tilde{x} > x_k^{(w)*} \\
\Rightarrow\ &x_k^{(v)*} - 2\eta(q+1)x_k^{(u)*} < (1 - 2\eta(q+1))x_k^{(w)*} \\
\Rightarrow\ &x_k^{(v)*} - x_k^{(w)*} < 2\eta(q+1)(x_k^{(u)*} - x_k^{(w)*}) \\
\Rightarrow\ &\eta > \frac{1}{2(q+1)}(\frac{x_k^{(v)*} - x_k^{(w)*}}{x_k^{(u)*} - x_k^{(w)*}})
\end{aligned}
\tag{13}
$$

Recall that $q$ is an integer and $q \geq 0$, thus

$$
1 - 2\eta(q+1) < 0 \Rightarrow \eta > \frac{1}{2(q+1)}
\tag{14}
$$

Also notice that

$$
x_k^{(w)*} < x_k^{(u)*} < x_k^{(v)*} \Rightarrow \frac{x_k^{(v)*} - x_k^{(w)*}}{x_k^{(u)*} - x_k^{(w)*}} > 1
\tag{15}
$$

Putting Inequalities 13, 14 and 15 together, we obtain $\eta > \frac{1}{2(q+1)}$.

**Case 2:** here $w, u, v$ are from right to left, and $x_k^{(w)*} > x_k^{(u)*} > x_k^{(v)*}$. This is very similar to Case 1, so we omit the details and provide the final result as

$$\eta > \frac{1}{2(q+1)}\left(\frac{x_k^{(w)*} - x_k^{(v)*}}{x_k^{(u)*} - x_k^{(v)*}}\right) \quad \text{and} \quad \frac{x_k^{(w)*} - x_k^{(v)*}}{x_k^{(u)*} - x_k^{(v)*}} > 1 \tag{16}$$

We will arrive at the same result: $\eta > \frac{1}{2(q+1)}$.

Note that the length of the pattern $x$ is $l$ and the length of input $y$ is $n$. As a result, the expected number of elements in $y$ aligned to a single $x_i$, $i \in [0, l-1]$ should be $n/l$, i.e. $\mathbb{E}[q] = n/l - 1$. Taking expectation on both sides of the above inequality, we obtain $\mathbb{E}[\eta] > \frac{1}{2(n/l-1+1)} = \frac{l}{2n}$. $\quad \square$

**Corollary 1.** *With the same assumption of Theorem 1, let pattern $x$ and input $y$ have the same length, i.e., $n = l$. In order to make one step jumping out of local region $u$, we should have $\mathbb{E}[\eta] > \frac{1}{2}$.*

## B Detailed Experimental Results for Multivariate DTWNet

Multivariate DTWs are often computed in two forms: MDTW-I and MDTW-D [21]. MDTW-I treats each dimension independently, so it is simply a stack of multiple univariate DTWs, thus directly applies to our method. MDTW-D needs to compute multivariate distance $\text{mdtw}^2 = \sum \|\mathbf{x_i} - \mathbf{y_j}\|^2, \mathbf{x_i} \in \mathbf{R^m}, \mathbf{y_j} \in \mathbf{R^m}$ in the Dynamic Programming step, instead of the scaler version $\sum \|x_i - y_j\|^2$. As long as the norm is well defined, e.g., Euclidean distance, the forward pass and the backpropagation are performed in the same manner. We can even define other distances, as long as their gradients w.r.t. to the vector $\mathbf{x}$ can be computed.

We run a 3-dim multivariate time series classification task here, using MDTW-D and Euclidean distance in our approach. The experiment settings follow Section 6.1 in the paper. The following figures show: one sample data (3-variate series) from each category, the learned kernel, test loss and test acc comparison. Our method (DTW) outperforms others.

(a) Data sample type-0     (b) Data sample type-1     (c) The learned kernel

(d) Test Loss           (e) Test Acc

Figure 6: Multivariate DTWNet Experiment

## C Detailed Experimental Results for Barycenter Experiment

Table 2: Barycenter Experiment, Average DTW Loss on Training Set

| | SoftDTW $\gamma = 1$ | SoftDTW $\gamma = 0.1$ | SoftDTW $\gamma = 0.01$ | SoftDTW $\gamma = 0.001$ | SSG | DBA | **Ours** |
|---|---|---|---|---|---|---|---|
| 50words | 7.337 | 5.294 | 5.375 | 5.478 | 4.904 | 4.604 | **4.415** |
| Adiac | 0.246 | 0.218 | 0.219 | 0.663 | **0.133** | 0.136 | 0.461 |
| ArrowHead | 2.713 | 2.223 | 1.690 | 1.979 | 1.856 | **1.555** | 1.717 |
| Beef | 17.753 | 9.260 | 7.889 | 8.617 | 17.056 | 8.618 | **7.487** |
| BeetleFly | 34.656 | 23.368 | 22.839 | 23.729 | 24.771 | 22.341 | **20.544** |
| BirdChicken | 21.017 | 10.711 | 9.459 | 11.982 | 11.115 | 12.768 | **9.136** |
| CBF | 22.421 | 14.292 | 12.891 | 14.062 | 11.086 | **11.039** | 11.592 |
| Car | 2.003 | 1.252 | 1.063 | 1.386 | **0.940** | 1.089 | 1.230 |
| ChlorineConcentration | 24.167 | 14.339 | 16.069 | 15.773 | **12.871** | 13.227 | 13.829 |
| CinC_ECG_torso | 137.172 | 99.014 | 80.749 | 91.645 | 80.430 | 79.076 | **69.681** |
| Coffee | 1.056 | 0.740 | 1.228 | 2.038 | **0.451** | 0.478 | 0.945 |
| Computers | 192.741 | 198.368 | 162.193 | 163.533 | **155.350** | 164.252 | 158.226 |
| Cricket_X | 45.766 | 33.312 | 32.477 | 33.539 | 33.537 | 32.728 | **30.582** |
| Cricket_Y | 43.959 | 31.407 | 31.865 | 30.490 | 31.170 | 32.640 | **29.766** |
| Cricket_Z | 48.030 | 34.976 | 34.390 | 35.690 | 33.936 | 34.711 | **31.129** |
| DiatomSizeReduction | 0.161 | 0.146 | 0.139 | 0.576 | 0.061 | **0.055** | 0.760 |
| DistalPhalanxOutlineAgeGroup | 1.791 | 1.366 | 1.573 | 2.406 | **1.088** | 1.141 | 1.332 |
| DistalPhalanxOutlineCorrect | 2.643 | 2.081 | 2.227 | 2.453 | 1.896 | **1.842** | 1.874 |
| DistalPhalanxTW | 1.358 | 1.010 | 1.031 | 1.399 | **0.691** | 0.736 | 0.937 |
| ECG200 | 7.365 | 5.677 | 7.488 | 6.931 | 6.171 | 6.262 | **5.638** |
| ECG5000 | 12.282 | 11.623 | 13.441 | 11.390 | 12.471 | 11.427 | **10.567** |
| ECGFiveDays | 9.987 | 8.418 | 7.821 | 7.001 | 7.750 | **6.927** | 11.559 |
| Earthquakes | 150.271 | **87.765** | 91.644 | 88.571 | 88.961 | 88.290 | 89.370 |
| ElectricDevices | 31.243 | 28.281 | 28.409 | 27.694 | 27.151 | 27.649 | **27.033** |
| FISH | 0.913 | 0.777 | 0.639 | 0.692 | **0.487** | 0.497 | 0.789 |
| FaceAll | 18.250 | 15.273 | 17.053 | 15.958 | 13.898 | 14.416 | **13.678** |
| FaceFour | 28.604 | 24.447 | 26.018 | 27.347 | 30.171 | 29.341 | **22.983** |
| FacesUCR | 16.913 | 14.094 | 15.200 | 15.657 | 12.979 | **12.656** | 13.025 |
| FordA | 63.812 | 53.893 | 55.744 | 55.351 | 51.829 | 53.025 | **51.076** |
| FordB | 66.695 | 56.032 | 56.154 | 55.071 | 52.866 | 53.447 | **51.686** |
| Gun_Point | 7.586 | 2.525 | 2.208 | 2.354 | 3.393 | **2.113** | 2.259 |
| Ham | 25.753 | 21.942 | 19.811 | 20.530 | 19.482 | 20.669 | **19.319** |
| HandOutlines | - | - | - | - | 2.094 | 1.975 | 2.859 |
| Haptics | 19.904 | 15.475 | 15.414 | 16.633 | 12.714 | 14.710 | 14.638 |
| Herring | 1.778 | 1.245 | 1.520 | 1.526 | **0.873** | 1.118 | 1.172 |
| InlineSkate | 91.103 | 34.482 | 25.691 | 27.585 | 30.498 | 22.163 | **21.671** |
| InsectWingbeatSound | 14.798 | 13.418 | 13.024 | 12.506 | 12.148 | 12.407 | **11.909** |
| ItalyPowerDemand | 2.748 | 2.317 | 3.343 | 2.810 | 2.222 | **2.161** | 2.302 |
| LargeKitchenAppliances | 125.113 | **99.668** | 104.094 | 100.850 | 109.575 | 112.017 | 102.100 |
| Lighting2 | 84.514 | 73.412 | 75.298 | 74.958 | 72.800 | **71.848** | 72.641 |
| Lighting7 | 32.775 | 27.390 | 25.715 | 26.188 | 25.786 | 25.216 | **24.230** |
| MALLAT | 4.744 | 5.130 | 3.332 | 4.122 | 2.091 | **1.949** | 3.461 |
| Meat | 0.820 | 0.492 | 1.046 | 1.122 | 0.040 | **0.039** | 2.381 |
| MedicalImages | 8.110 | 8.041 | 8.662 | 9.552 | **6.188** | 6.900 | 6.477 |
| MiddlePhalanxOutlineAgeGroup | 0.853 | 0.748 | 0.914 | 1.097 | **0.511** | 0.511 | 0.766 |
| MiddlePhalanxOutlineCorrect | 0.825 | 0.652 | 0.720 | 1.181 | **0.507** | 0.508 | 0.551 |
| MiddlePhalanxTW | 0.752 | 0.584 | 0.949 | 1.449 | 0.416 | **0.404** | 0.692 |
| MoteStrain | 24.706 | 22.632 | **19.159** | 20.079 | 21.629 | 20.790 | 20.155 |
| NonInvasiveFatalECG_Thorax1 | 2.418 | 2.607 | 6.163 | 3.169 | **1.139** | 1.149 | 2.852 |
| NonInvasiveFatalECG_Thorax2 | 2.311 | 2.169 | 2.536 | 2.858 | 1.088 | **1.081** | 2.318 |
| OSULeaf | 32.206 | 21.743 | 21.641 | 20.844 | 20.371 | 19.971 | **19.120** |
| OliveOil | 1.217 | 1.180 | 1.362 | 4.419 | 0.018 | **0.017** | 1.187 |
| PhalangesOutlinesCorrect | 1.681 | 1.312 | 1.946 | 1.684 | **1.132** | 1.146 | 1.277 |
| Phoneme | 181.389 | 134.930 | 135.640 | 136.674 | 133.475 | 121.774 | **118.926** |
| Plane | 1.058 | 0.783 | 1.202 | 1.809 | 0.444 | **0.416** | 1.075 |
| ProximalPhalanxOutlineAgeGroup | 0.620 | 0.530 | 0.807 | 0.946 | 0.352 | **0.336** | 0.431 |
| ProximalPhalanxOutlineCorrect | 0.859 | 0.707 | 1.053 | 1.164 | **0.512** | 0.514 | 0.537 |
| ProximalPhalanxTW | 0.630 | 0.513 | 0.645 | 1.089 | **0.247** | 0.261 | 0.800 |
| RefrigerationDevices | 182.659 | 156.169 | 151.285 | 149.829 | 152.254 | 155.795 | **137.791** |
| ScreenType | 187.460 | 156.534 | 155.496 | 155.359 | **151.867** | 158.273 | 153.550 |
| ShapeletSim | 236.166 | 123.055 | 124.282 | 127.652 | 122.746 | 123.699 | **111.356** |
| ShapesAll | 15.045 | 8.828 | 7.745 | 8.333 | 8.755 | 8.929 | **7.448** |
| SmallKitchenAppliances | 184.888 | 177.515 | 181.719 | **177.369** | 177.863 | 181.845 | 179.879 |
| SonyAIBORobotSurface | 8.765 | 6.722 | 7.511 | 7.685 | **5.876** | 6.252 | 6.663 |
| SonyAIBORobotSurfaceII | 11.896 | 10.897 | 12.207 | 12.315 | **9.463** | 9.754 | 11.239 |
| StarLightCurves | 16.522 | 9.449 | 6.834 | 6.645 | 6.557 | **6.156** | 6.448 |
| Strawberry | 2.034 | 1.642 | 1.315 | 1.704 | 1.225 | **1.223** | 1.489 |
| SwedishLeaf | 2.891 | 2.124 | 2.419 | 2.400 | **1.887** | 1.936 | 2.149 |
| Symbols | 2.140 | 1.012 | **0.798** | 1.212 | 1.133 | 0.882 | 1.573 |
| ToeSegmentation1 | 35.839 | 29.387 | 27.285 | 26.861 | 27.618 | 30.725 | **26.201** |
| ToeSegmentation2 | 36.316 | 26.012 | 23.562 | 22.410 | 24.837 | 24.226 | **21.764** |
| Trace | 2.169 | 2.206 | 1.249 | 1.374 | **0.767** | 0.964 | 0.981 |
| TwoLeadECG | 1.616 | 1.354 | 1.900 | 1.349 | **1.015** | 1.118 | 1.084 |
| Two_Patterns | 12.811 | 10.047 | 8.079 | **7.850** | 8.528 | 9.718 | 8.010 |
| UWaveGestureLibraryAll | 77.858 | 46.754 | 43.451 | 42.883 | 42.470 | 43.537 | **37.055** |
| Wine | 0.738 | 0.517 | 0.633 | 1.109 | **0.112** | 0.113 | 0.435 |
| WordsSynonyms | 17.235 | 11.125 | 10.347 | 10.832 | 12.036 | 11.489 | **8.428** |
| Worms | 107.089 | 73.280 | 68.799 | 74.002 | 75.779 | 80.812 | **60.592** |
| WormsTwoClass | 128.791 | 86.165 | 82.996 | 82.916 | 83.880 | 90.286 | **77.192** |
| synthetic_control | 16.805 | 8.880 | 8.886 | 9.295 | 8.924 | 8.926 | **8.328** |
| uWaveGestureLibrary_X | 33.867 | 20.419 | 19.155 | 19.131 | 18.673 | 19.393 | **17.997** |
| uWaveGestureLibrary_Y | 35.155 | 19.121 | 16.982 | 17.484 | 18.505 | 16.598 | **15.786** |
| uWaveGestureLibrary_Z | 33.574 | 19.668 | 18.401 | 18.701 | 18.693 | 18.248 | **16.934** |
| wafer | 30.883 | **21.369** | 24.101 | 25.974 | 23.579 | 31.298 | 24.725 |
| yoga | 33.428 | 14.453 | 14.055 | 11.822 | 11.343 | 12.424 | **10.882** |

Table 3: Barycenter Experiment, Average DTW Loss on Testing Set

| | SoftDTW $\gamma = 1$ | SoftDTW $\gamma = 0.1$ | SoftDTW $\gamma = 0.01$ | SoftDTW $\gamma = 0.001$ | SSG | DBA | Ours |
|---|---|---|---|---|---|---|---|
| 50words | 13.770 | 11.642 | 11.031 | 10.913 | 11.162 | 11.195 | **10.709** |
| Adiac | 0.838 | 0.305 | 0.285 | 0.585 | **0.245** | 0.256 | 0.564 |
| ArrowHead | 5.427 | 4.402 | 3.690 | **3.478** | 3.766 | 3.688 | 4.009 |
| Beef | 14.946 | 14.178 | 9.814 | 11.599 | 14.822 | 12.016 | **8.813** |
| BeetleFly | 53.030 | 38.986 | **36.598** | 38.443 | 41.506 | 40.907 | 36.863 |
| BirdChicken | 43.657 | **19.897** | 22.499 | 20.594 | 36.865 | 30.522 | 22.585 |
| CBF | 24.343 | 15.210 | 17.237 | 15.566 | 14.194 | 14.596 | **13.755** |
| Car | 4.355 | 3.072 | 2.723 | 2.753 | 2.654 | **2.462** | 2.573 |
| ChlorineConcentration | 25.474 | 17.345 | 18.659 | 17.919 | **16.459** | 17.341 | 16.492 |
| CinC_ECG_torso | 180.663 | 141.548 | 141.753 | 132.553 | 166.838 | 136.126 | **128.023** |
| Coffee | 1.226 | 0.826 | 1.293 | 1.577 | 0.658 | **0.654** | 1.045 |
| Computers | 176.868 | 157.962 | 148.396 | 153.658 | **146.668** | 153.739 | 159.314 |
| Cricket_X | 51.207 | 37.780 | 38.551 | 37.916 | 36.245 | 37.398 | **36.214** |
| Cricket_Y | 43.451 | 34.059 | 33.094 | 33.679 | **32.414** | 33.642 | 32.629 |
| Cricket_Z | 48.466 | 36.712 | 36.351 | 37.211 | 36.525 | 36.586 | **35.056** |
| DiatomSizeReduction | 4.010 | 4.024 | 3.902 | **3.896** | 3.899 | 3.907 | 4.246 |
| DistalPhalanxOutlineAgeGroup | 1.523 | 1.304 | 1.526 | 2.036 | **1.061** | 1.065 | 1.841 |
| DistalPhalanxOutlineCorrect | 2.785 | 2.351 | 2.750 | 2.557 | 1.958 | **1.956** | 1.979 |
| DistalPhalanxTW | 1.405 | 0.997 | 1.977 | 1.705 | 0.806 | **0.806** | 1.179 |
| ECG200 | 9.344 | 7.555 | 8.911 | 8.784 | 7.780 | 8.422 | **6.959** |
| ECG5000 | 26.295 | 22.558 | 22.592 | 24.241 | 25.302 | 27.389 | **22.372** |
| ECGFiveDays | 8.767 | 10.175 | 10.589 | 10.965 | 7.422 | **7.336** | 8.647 |
| Earthquakes | 156.759 | 110.492 | 108.953 | 109.385 | 106.678 | **102.084** | 107.288 |
| ElectricDevices | 43.261 | 37.808 | 37.130 | 37.421 | 36.852 | 36.939 | **35.069** |
| FISH | 1.808 | 1.661 | 1.535 | 1.682 | 1.464 | **1.422** | 1.652 |
| FaceAll | 20.162 | 18.590 | 19.029 | 20.625 | 17.536 | **17.428** | 19.027 |
| FaceFour | 38.929 | 38.416 | 40.709 | 40.024 | **36.045** | 39.897 | 39.898 |
| FacesUCR | 20.339 | 19.555 | 20.671 | 20.296 | 17.635 | **17.090** | 18.105 |
| FordA | 65.742 | 56.997 | 55.973 | 55.459 | 53.513 | 53.965 | **52.667** |
| FordB | 69.145 | 60.406 | 61.231 | 59.489 | 57.921 | 58.954 | **56.693** |
| Gun_Point | 8.030 | 2.816 | 2.478 | 2.428 | 3.172 | 2.796 | **2.315** |
| Ham | 31.478 | 28.953 | 26.317 | 30.149 | 26.639 | 26.455 | **25.510** |
| HandOutlines | - | - | - | - | 10.864 | 10.794 | **7.733** |
| Haptics | 25.431 | 22.077 | 20.684 | 21.966 | **17.196** | 17.926 | 24.014 |
| Herring | 1.465 | 1.385 | 1.331 | 1.462 | **0.915** | 0.948 | 1.372 |
| InlineSkate | 127.648 | 65.537 | 48.215 | 45.301 | 58.181 | 55.270 | **40.608** |
| InsectWingbeatSound | 17.576 | 16.680 | 15.102 | **15.010** | 16.210 | 15.079 | 15.493 |
| ItalyPowerDemand | 2.523 | 2.578 | 2.523 | 2.820 | **2.147** | 2.705 | 2.369 |
| LargeKitchenAppliances | 118.456 | 114.174 | **107.120** | 107.325 | 122.776 | 131.369 | 110.499 |
| Lighting2 | 86.450 | 75.184 | **72.139** | 81.725 | 78.196 | 78.350 | 73.070 |
| Lighting7 | 48.326 | 37.158 | **36.557** | 38.376 | 37.673 | 37.747 | 41.663 |
| MALLAT | 6.002 | 4.692 | 4.717 | 6.280 | 3.668 | **3.379** | 4.938 |
| Meat | 0.593 | 0.259 | 0.438 | 1.569 | 0.041 | **0.041** | 0.470 |
| MedicalImages | 8.252 | 8.807 | 9.334 | 8.741 | 7.194 | **6.916** | 7.762 |
| MiddlePhalanxOutlineAgeGroup | 0.835 | 0.770 | 1.540 | 1.158 | **0.723** | 0.733 | 0.962 |
| MiddlePhalanxOutlineCorrect | 1.235 | 1.223 | 1.145 | 2.254 | 1.187 | 1.200 | **0.998** |
| MiddlePhalanxTW | 0.836 | 0.729 | 1.054 | 1.051 | 0.592 | **0.570** | 0.694 |
| MoteStrain | 22.970 | 23.938 | 22.080 | 24.643 | 21.964 | **21.094** | 25.032 |
| NonInvasiveFatalECG_Thorax1 | 2.745 | 2.919 | 3.560 | 3.890 | 1.548 | **1.509** | 3.561 |
| NonInvasiveFatalECG_Thorax2 | 2.384 | 2.888 | 3.146 | 3.711 | **1.465** | 1.521 | 2.955 |
| OSULeaf | 35.193 | 26.392 | 23.544 | 24.177 | 25.619 | **23.188** | 23.352 |
| OliveOil | 0.961 | 0.747 | 2.107 | 2.002 | 0.020 | **0.020** | 1.082 |
| PhalangesOutlinesCorrect | 2.067 | 1.556 | 1.782 | 1.487 | 1.254 | **1.239** | 1.251 |
| Phoneme | 315.058 | 291.260 | **286.486** | 286.661 | 319.069 | 311.551 | 289.715 |
| Plane | 1.171 | 0.752 | 1.247 | 1.572 | **0.534** | 0.553 | 1.109 |
| ProximalPhalanxOutlineAgeGroup | 0.595 | 0.461 | 0.735 | 1.230 | 0.356 | **0.351** | 0.521 |
| ProximalPhalanxOutlineCorrect | 0.704 | 0.587 | 0.659 | 1.020 | **0.432** | 0.435 | 0.536 |
| ProximalPhalanxTW | 0.775 | 0.586 | 0.707 | 1.699 | 0.343 | **0.348** | 0.815 |
| RefrigerationDevices | 198.045 | 167.040 | 162.204 | 164.149 | 164.174 | 174.493 | **156.454** |
| ScreenType | 143.046 | **119.853** | 126.123 | 124.484 | 123.647 | 136.127 | 135.216 |
| ShapeletSim | 243.103 | 150.826 | 151.325 | 152.915 | 153.454 | 153.326 | **146.383** |
| ShapesAll | 21.230 | 15.139 | 12.812 | 12.824 | 14.086 | 14.084 | **12.408** |
| SmallKitchenAppliances | 176.407 | 173.053 | 175.462 | **171.829** | 178.142 | 173.198 | 181.316 |
| SonyAIBORobotSurface | 8.647 | 9.489 | 9.221 | 10.841 | 7.459 | **7.430** | 7.882 |
| SonyAIBORobotSurfaceII | 16.137 | 17.066 | 16.947 | 17.336 | **14.439** | 15.585 | 15.215 |
| StarLightCurves | 17.915 | 10.925 | 7.939 | 7.484 | 7.457 | 7.376 | **7.316** |
| Strawberry | 2.706 | 1.921 | 2.757 | 2.635 | 1.609 | 1.656 | **1.599** |
| SwedishLeaf | 2.842 | 2.442 | 2.483 | 2.688 | 2.087 | **2.073** | 2.395 |
| Symbols | 6.280 | 5.245 | 3.973 | **3.930** | 5.395 | 4.862 | 4.498 |
| ToeSegmentation1 | 43.606 | 36.174 | 34.410 | 35.602 | 35.703 | 36.982 | **34.158** |
| ToeSegmentation2 | 72.831 | 58.728 | **47.650** | 51.515 | 54.558 | 57.188 | 53.725 |
| Trace | 2.879 | 1.720 | 1.374 | 1.448 | **0.818** | 0.963 | 1.048 |
| TwoLeadECG | 1.619 | 1.275 | 1.441 | 1.708 | **1.185** | 1.238 | 1.383 |
| Two_Patterns | 12.546 | 9.490 | 8.107 | **8.012** | 9.502 | 8.415 | 8.943 |
| UWaveGestureLibraryAll | 78.820 | 51.054 | 46.568 | 47.568 | 45.660 | 47.290 | **41.052** |
| Wine | 0.807 | 0.549 | 2.339 | 1.202 | 0.103 | **0.101** | 0.765 |
| WordsSynonyms | 25.751 | 18.486 | 17.173 | 16.387 | 20.113 | 19.359 | **16.247** |
| Worms | 169.897 | 111.593 | 99.132 | 97.843 | 123.028 | 111.755 | **94.781** |
| WormsTwoClass | 172.790 | 119.187 | 110.483 | **105.179** | 114.444 | 113.869 | 105.696 |
| synthetic_control | 17.031 | 9.794 | 9.631 | 9.806 | 9.620 | 9.692 | **9.126** |
| uWaveGestureLibrary_X | 34.245 | 21.273 | 19.248 | 19.091 | 20.334 | 20.586 | **18.738** |
| uWaveGestureLibrary_Y | 37.343 | 20.565 | 19.212 | 18.875 | 19.129 | 19.481 | **17.015** |
| uWaveGestureLibrary_Z | 35.221 | 22.716 | 20.444 | 20.108 | 20.072 | 21.296 | **18.303** |
| wafer | 30.642 | **23.935** | 27.221 | 24.758 | 30.820 | 32.328 | 24.577 |
| yoga | 24.982 | 14.573 | 15.286 | 11.849 | 11.681 | 11.887 | **11.120** |