[Reviews · NeurIPS 2019]

Reviewer 1



The method is novel, the work is well done and presents high-quality research. The presentation is excellent, the method is explained in a clear manner and with sufficient details to reproduce the results (although not needed as the authors provide the code). The presented work is significant, with both academic and industrial impact.

Reviewer 2



There are many papers that have tried to combine DTW and neural networks, but the alternating optimization approach in this paper appears to be new and significant to me. While the idea of the paper is exciting, the execution is poor and the writing is disappointing. Here are the main feedback points: 1. The writing omits the description of many notations. For example, I could not find the definition of the forward pass for the main network G_{x, w}. I could not figure out how they go from x to z. Another example is in line 4 of Algorithm 1. It refers to f_t(x, y) in Eq. (2), but it is not found there. I did search the provided code for answers to the above questions, but didn't spend much time reading because it lacked a proper documentation. 2. It is possible to prove some form of convergence results for alternating optimization algorithms. However, the theoretical analysis in this paper is not rigorous. In particular, Theorem 1 makes unacceptable assumptions. For example, consider "Assume that x ̃ is close to global minima": This assumption may never hold. We cannot put assumptions on the solutions, we only can put assumptions on the initial conditions and the learning algorithms and then show that the solution will be in the \epsilon neighborhood of the final solutions after t iterations. 3. It seems that the idea only will work for the univariate time series. Can the authors comment on this? 4. The experiments are insufficient. The authors need to evaluate their proposed algorithm on more and diverse datasets and report the speed results. One dataset from UCR repo is not enough. 5. The authors have some handwaving claims such as interpretability in the abstract without fully supporting them in the experiments. === After reading the response and discussions === I appreciate the authors response. I keep my score. The authors argument about non-linearity of neural network and thus having low-quality theorems is not acceptable. These days there are plenty of high-quality theorems for deep learning. The authors argument about interpretability is also invalid. Overall, I don't think this paper is ready for publication. === Update 2 After discussion with Reviewer #1, I updated my vote. The authors need to seriously improve the presentation of this paper.

Reviewer 3



This paper presents a learning framework based on Dynamic Time Warping. DTW kernel is applied in neural networks with stochastic backpropagation. Both end-ot-end DTW and streaming version are implemented in this work. The method has been tested on both synthetic and real data. This work applies learnable DTW kernels in neural networks to represent Doppler invariance in the data. In instead of using DTW as a loss function, DTW kernel is employed obtain a better feature extraction and generates interpretable representation. DTW loss and convergence is theoretically analysed. The proposed technique can be further improved. For example, it will be helpful to know how to decide the number of DTW layers. In algorithm 1’s INPUT part, kernels are initially set as input. However the kernels are randomly initialised in OUTPUT part, which should be clarified. Also, in the experiment, it will be helpful if the proposed method can be tested with more real datasets for the application part.

[Author Response · NeurIPS 2019]

We thank all the reviewers for their valuable comments. We carefully address all the raised issues accordingly below.

**1    Response to Reviewer 1:** We appreciate your positive feedback.

**2    Response to Reviewer 2:**

Thank you for the comments. We will proofread and improve the readability. Please see details below.

**2.1    Q: Definition of the forward pass for the main network $G_{x,w}$. How they go from x to z.**
A: $G_{x,w}$ is the network parameterized by the kernel $x$ and linear layers' weights $w$. Input is $y$, and output $z = G_{x,w}(y)$.

**2.2    Q: In line 4 of Algorithm 1, it refers to $f_t(x,y)$ in Eq. (2), but it is not found there.**
A: Sorry for the confusion. $f_t(x,y)$ is the same as $dtw^2(x,y)$ in Eq.2, with the subscript $t$ being the iteration index.

**2.3    Q: "Assume that $\tilde{x}$ is close to global minima". This assumption may never hold.**
A: We shouldn't abuse the term "assumption". In fact, this "assumption" is not necessary for the analysis. We simply
want to emphasize that we are interested in areas having many local minima, which happen to be always close to the
global minimum from empirical observations. Note that the local-minima areas of interests can be anywhere, and the
analysis still holds. The only issue arises when the left region of $w$'s stationary point $x^{(w)*}$ is located to the right of
region $w$. In this case, we can combine $w$ and the center region $u$ to form a new quasi-bowl center region $u' = [w, u]$
(similar to the analysis at the top of Page 6 in the paper), and the analysis still holds. We will remove this "assumption".

**2.4    Q: Proof of convergence: after t iterations, it will be eps-close to the exact DTW.**
Since it is highly non-convex and non-smooth, to the best of our knowledge, without strong assumptions it is unlikely to
prove global convergence. We also analyze the escaping behavior (from local minima) but not the global convergence.
The shape of DTW loss is identified through the alternating algorithm, which is one of the contributions in our paper.

**2.5    Q: The idea will only work for the univariate time series.**
A: The proposed approach works for both univariate and multivariate time series. Multivariate DTWs are often
computed in two forms: MDTW-I and MDTW-D [1]. MDTW-I treats each dimension independently, so it is simply
a stack of multiple univariate DTWs, thus directly applies to our method. MDTW-D needs to compute multivariate
distance $||\mathbf{x_i} - \mathbf{y_j}||^2, \mathbf{x_i} \in \mathbf{R^m}, \mathbf{y_j} \in \mathbf{R^m}$ in the Dynamic Programming step, instead of the scaler version $||x_i - y_j||^2$.
As long as the norm is well defined, e.g., Euclidean distance, the forward pass and the backpropagation are performed
in the same manner. We can even define other distances, as long as their gradients w.r.t. to $\mathbf{x}$ can be computed.

We run a 3-dim multivariate time series classification task here, using MDTW-D and Euclidean distance in our approach.
The experiment settings follow Section 6.1 in the paper. The following figures show: one sample data from each
category, the learned kernel, test loss and test acc comparison. Our method (DTW) outperforms others.

(a) Data sample type-0  (b) Data sample type-1 | (c) The learned kernel    (d) Test Loss    (e) Test Acc

**2.6    Q: The experiments are insufficient. One dataset from UCR repo is not enough.**
A: The UCR repo is a collection of a large variety of time series data, being the standard benchmark repo in related
publications. We performed comprehensive experiments on 85 datasets from UCR repo (details provided in appendix
of the paper), to make a fair comparison with Soft-dtw (they only use the UCR repo in their original paper as well).
Additional experiments on synthetic data are also carried out. But we agree that more datasets should be considered in
this area and we are exploring them in the final version and subsequent work.

**2.7    Q: Some handwaving claims such as interpretability.**
A: Some interpretability is revealed by the learned kernel's shape matching true patterns in the data (Fig 4, 5 in paper).

**3    Response to Reviewer 3:** Thank you for your comments and please see details below.

**3.1    Q: For example, it will be helpful to know how to decide the number of DTW layers.**
A: Empirically speaking, 1 or 2 layers of DTW are good enough. We observe no clear improvement with more than 2
layers, but the model complexity would be increased dramatically.

**3.2    In alg 1's INPUT, kernels are initially set as input. But they are randomly initialised in OUTPUT part.**
A: Kernels $x$ and weights $w$ are not inputs but model parameters being randomly initialized. We will clarify it.

**3.3    Q: It will be helpful if the proposed method can be tested with more real datasets for the application part.**
A: We agree with testing more datasets. Due to page limit, we randomly select the Haptics dataset from UCR repo as an
expressive example in the application section, but more datasets will be considered (please also see response 2.6).

[1] Mohammad Shokoohi-Yekta, Bing Hu, Hongxia Jin, Jun Wang, and Eamonn Keogh. Generalizing dtw to the
multi-dimensional case requires an adaptive approach. *Data mining and knowledge discovery*, 31(1):1–31, 2017.


[Meta-Review · NeurIPS 2019]

The paper proposes an approach for incorporating Dynamic Time Warping kernels in a neural network. The method is shown to perform well on both synthetic and real data. The reviewers think that this is a novel and potentially impactful contribution to the community. The concerns raised by the reviewers were successfully addressed by the authors or clarified during the discussion. We encourage the authors to improve the presentation of the paper and to add the results as indicated in the rebuttal.